# ZERO-SHOT DUAL MACHINE TRANSLATION

## ABSTRACT

Neural Machine Translation (NMT) systems rely on large amounts of parallel data. This is a major challenge for low-resource languages. Building on recent work on unsupervised and semi-supervised methods, we present an approach that combines zero-shot and dual learning. The latter relies on reinforcement learning, to exploit the duality of the machine translation task, and requires only monolingual data for the target language pair. Experiments on the UN corpus show that a zero-shot dual system, trained on English-French and English-Spanish, outperforms by large margins a standard NMT system in zero-shot translation performance on Spanish-French (both directions). We also evaluate on *newstest2014*. These experiments show that the zero-shot dual method outperforms the LSTM-based unsupervised NMT system proposed in (Lample et al., 2018b), on the en→fr task, while on the fr→en task it outperforms both the LSTM-based and the Transformers-based unsupervised NMT systems.

## 1 INTRODUCTION

The availability of large high-quality parallel corpora is key to building robust machine translation systems. Obtaining such training data is often expensive or outright infeasible, thereby constraining the availability of translation systems, especially for low-resource language pairs. Among the work trying to alleviate this data-bottleneck problem, Johnson et al. (2016) present a system that performs *zero-shot* translation. Their approach is based on a multi-task model trained to translate between multiple language pairs with a single encoder-decoder architecture. Multi-tasking is accomplished by a special symbol, introduced at the beginning of the source sentence, that identifies the desired target language. Their idea enables the NMT model of Wu et al. (2016) to translate between language pairs not encountered during training, as long as source and target language were part of the training data.

Since any machine translation task has a canonical inverse problem, namely that of translating in the opposite direction, He et al. (2016) propose a dual learning method for NMT. They report improved translation quality of two dual, weakly trained translation models using *monolingual* data from both languages. For that, a monolingual sentence is translated twice, first into the target language and then back into the source language. A language model is used to measure the *fluency* of the intermediate translation. In addition, a *reconstruction* error is computed via back-translation. Fluency and reconstruction scores are combined in a reward function which is optimized with reinforcement learning.

We propose an approach that builds upon the multilingual NMT architecture of (Johnson et al., 2016) to improve the zero-shot translation quality by applying the reinforcement learning ideas from the dual learning work (He et al., 2016). Our model outperforms the zero-shot system's performance while being simpler and more scalable than the original dual formulation, as it learns a single model for all involved translation directions. Moreover, for the original dual method to work, some amount of parallel data is needed in order to train an initial weak translation model. If no parallel data is available for that language pair, the original dual approach is not feasible, as a random translation model is ineffective in guiding exploration for reinforcement learning. In our formulation, however, no parallel data is required at all to make the algorithm work, as the zero-shot mechanism kick-starts the dual learning.

Experiments show that our approach outperforms by large margins the zero-shot translation of the NMT model using only monolingual corpora. Finally, we prove that zero-shot dual NMT is competitive with recent unsupervised methods such as (Lample et al., 2018b).

## 2 RELATED WORK

We build on the standard encoder-attention-decoder architecture for neural machine translation (Sutskever et al., 2014; Cho et al., 2014; Bahdanau et al., 2015), and specifically on the Google NMT implementation (Wu et al., 2016). This architecture has been extended by Johnson et al. (2016) for unsupervised or zero-shot translation. Here, one single multi-task system is trained on a subset of language pairs, but then can be used to translate between all pairs, including unseen ones. The success of this approach will depend on the ability to learn interlingua semantic representations of which some evidence is provided in (Johnson et al., 2016). In a zero-shot evaluation from Portuguese to Spanish Johnson et al. (2016) report a gap of 6.75 BLEU points from the supervised NMT model.

He et al. (2016) propose a learning mechanism that bootstraps an initial low-quality translation model. This so-called dual learning approach leverages criteria to measure the fluency of the translation and the reconstruction error of back-translating into the source language. As shown in (He et al., 2016), this can be effective, provided that the initial translation model provides a sufficiently good starting point. In the reported experiments, they bootstrap an NMT model (Bahdanau et al., 2015) trained on 1.2M parallel sentences. Dual learning improves the BLEU score on WMT'14 French-to-English by 5.23 points, obtaining comparable accuracy to a model trained on 10x more data, i.e. 12M parallel sentences.

Xia et al. (2017) use dual learning in a supervised setting and improve WMT'14 English-to-French translation by 2.07 BLEU points over the baseline of Bahdanau et al. (2015); Jean et al. (2014). In the supervised setting Bahdanau et al. (2017) show that Actor-Critic can be more effective than vanilla policy gradient (Williams, 1992).

Recent work has also explored fully unsupervised machine translation. Lample et al. (2018a) propose a model that takes sentences from monolingual corpora from both languages and maps them into the same latent space. Starting from an unsupervised word-by-word translation, they iteratively train encoder-decoder to reconstruct and translate from a noisy version of the input. Latent distributions from the two domains are aligned using adversarial discriminators. At test time, encoder and decoder work as a standard NMT system. Lample et al. (2018a) obtain BLEU score of 15.05 on WMT'14 English-to-French translation, improving over a word-by-word translation baseline with inferred bilingual dictionary (Conneau et al., 2018) by 8.77 BLEU points.

Artetxe et al. (2018) present a method related to Lample et al. (2018a). The model is an encoder-decoder system with attention trained to perform autoencoding and on-the-fly back-translation. The system handles both translation directions at the same time. Moreover, it shares the encoder across languages, in order to generate language independent representations, by exploiting pre-trained cross-lingual embeddings. This approach performs comparably on WMT'14 English-to-French, improving by 0.08 BLEU points over Lample et al. (2018a). Furthermore, WMT'14 French-to-English translation obtains 15.56 BLEU points.

Lample et al. (2018b) combine the previous two approaches. They first learn NMT language models, as denoising autoencoders, over the source and target languages. Leveraging these, two translation models are initialized, one in each direction. Iteratively, source and target sentences are translated using the current translation models and new translation models are trained on the generated pseudo-parallel sentences. The encoder-decoder parameters are shared across the two languages to share the latent representations and for regularization, respectively. Their results outperform those of Lample et al. (2018a); Artetxe et al. (2018), obtaining 25.1 BLEU points on WMT'14 English-to-French translation.

## 3 ZERO-SHOT DUAL MACHINE TRANSLATION

Our method for unsupervised machine translation works as follows: We start with a multi-language NMT model as used in zero-shot translation (Johnson et al., 2016). However, we then train this model using only monolingual data in a dual learning framework, similar to that proposed by He et al. (2016).

At least three languages are involved in our setting. Let them be X, Y and Z. Let the pairs Z-X and Z-Y be the language pairs with available parallel data and X-Y be the target language pair without. We thus assume to have access to sufficiently large parallel corpora $D_{Z \leftrightarrow X}$ and $D_{Z \leftrightarrow Y}$,

monolingual data $D_X$ and $D_Y$ for training the language models, as well as a small corpus $D_{X \leftrightarrow Y}$ for the low-resource pair evaluation.

## 3.1 ZERO-SHOT DUAL TRAINING

A single multilingual NMT model with parameters $\theta$ is trained on both $D_{Z \leftrightarrow X}$ and $D_{Z \leftrightarrow Y}$, in both directions. We stress that this model does not train on any X-Y sentence pairs. The multilingual NMT model is capable of generalizing to unseen pairs, that is, it can translate from X to Y and vice versa. Yet, the quality of the zero-shot pt→es translations, in the best setting (Model 2 in (Johnson et al., 2016)) is approximately 7 BLEU points lower than the corresponding supervised model.

Using the monolingual data $D_X$ and $D_Y$, we train two disjoint language models with maximum likelihood. We implement the language models as multi-layered LSTMs[1] that process one word at a time. For a given sentence, the language model outputs a probability $P_X(.)$, quantifying the degree of fluency of the sentence in the corresponding language X. Similar to He et al. (2016), we define a REINFORCE-like learning algorithm (Williams, 1992), where the translation model is the *policy* that we want to learn. The procedure on one sample is, schematically:

1. Given a sentence $\mathbf{x}$ in language X, sample a corresponding translation from the multilingual NMT model $\mathbf{y} \sim P_\theta(\cdot|\mathbf{x})$.
2. Compute the fluency reward, $r_1 = \log P_Y(\mathbf{y})$.
3. Compute the reconstruction reward $r_2 = \log P_\theta(\mathbf{x}|\mathbf{y})$ by using $P_\theta(\cdot|\mathbf{y})$ for back-translation.
4. Compute the total reward for $\mathbf{y}$ as $R = \alpha r_1 + (1 - \alpha) r_2$, where $\alpha$ is a hyper-parameter.
5. Update $\theta$ by scaling the gradient of the cross entropy loss by the reward $R$.

The process can be started with a sentence from either language X or Y and works symmetrically. Notice that this process captures the same main three principles that characterize the unsupervised approach of Lample et al. (2018b): Initialization, Language Modeling and Back-translation. However, rather than being combined in an algorithmic procedure, the reinforcement learning formulation allows to utilize them as components of a composite objective, which is optimized directly. This is a conceptually simple, flexible formulation. We envision that it may allow, among other things, the modeling of additional properties of good translations; as long as they can be implemented as reward functions.

## 3.2 DETAILS OF THE GRADIENT COMPUTATION

The reward for generating the target $\mathbf{y}$ from source $\mathbf{x}$ is defined as:

$$R(\mathbf{y}) = \alpha \log P_Y(\mathbf{y}) + (1 - \alpha) \log P_\theta(\mathbf{x}|\mathbf{y}) \tag{1}$$

We want to optimize the expected reward, under the translation model (the policy) $P_\theta$ and thus compute the gradient with respect to the expected reward:

$$\nabla_\theta \mathbb{E}_{\mathbf{y}|\mathbf{x}}[R] = \nabla_\theta \sum_{\mathbf{y}} P_\theta(\mathbf{y}|\mathbf{x}) R(\mathbf{y}) = \sum_{\mathbf{y}} \nabla_\theta P_\theta(\mathbf{y}|\mathbf{x}) R(\mathbf{y}) + P_\theta(\mathbf{y}|\mathbf{x}) \nabla_\theta R(\mathbf{y}) \tag{2}$$

$$= \mathbb{E}_{\mathbf{y}|\mathbf{x}} [R(\mathbf{y}) \nabla_\theta \log P_\theta(\mathbf{y}|\mathbf{x}) + (1 - \alpha) \nabla_\theta \log P_\theta(\mathbf{x}|\mathbf{y})] \tag{3}$$

Here we have used the product rule and the identity $\nabla P = P \cdot \nabla \log P$. Notice that, the reward depends (differentiably) on $\theta$, because of the reconstruction term.

Since the gradient is a conditional expectation, one can sample $\mathbf{y}$ to obtain an unbiased estimate of the expected gradient (Sutton et al., 2000):

$$\nabla_\theta \mathbb{E}_{\mathbf{y}|\mathbf{x}}[R] \approx \frac{1}{K} \sum_i R(\mathbf{y}_i) \nabla_\theta \log P_\theta(\mathbf{y}_i|\mathbf{x}) + (1 - \alpha) \nabla_\theta \log P_\theta(\mathbf{x}|\mathbf{y}_i) \tag{4}$$

---

[1]`https://www.tensorflow.org/tutorials/recurrent`

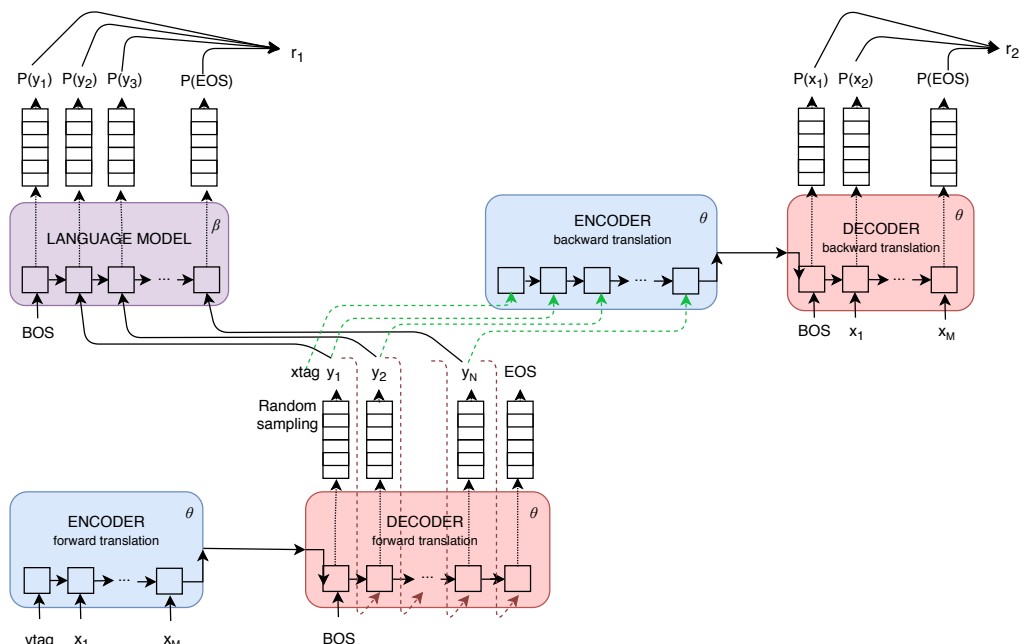

Figure 1: The zero-shot dual learning procedure (bottom-up, left-to-right). The encoder, on the forward translation step, receives the source sentence $\mathbf{x}$ as input with the target language tag $y_{tag}$. The decoder, on the forward translation step, randomly samples a translation $\mathbf{y}$, which is passed to the language model to generate the fluency reward. At the same time, $\mathbf{y}$ is encoded, by the same model, now with the source language tag $x_{tag}$. Then the decoder is used to compute the probability of the original sentence $\mathbf{x}$, conditioning on $\mathbf{y}$. Both encoders and decoders share the same parameters.

Any sampling approach can be used to estimate the expected gradient. He et al. (2016) use beam search instead of sampling to avoid large variance and unreasonable results. We generate from the policy by randomly sampling the next word from the predicted word distribution. However, we found it helpful to make the distribution more deterministic using a low softmax temperature (0.002). In order to reduce variance further, we compute a baseline through the batch. That is, we compute the average of each reward through the whole batch and subtract it from each reward. We optimize the likelihood of the translation system on the monolingual data of both the source and target languages, while scaling the gradients (equivalently, the loss or the negative likelihood) with the total rewards.

### 3.3 SYSTEM IMPLEMENTATION

The system is implemented as an extension of the Tensorflow (Abadi et al., 2016) NMT tutorial code[2], and is released as open source. Figure 1 visualizes the training process. The model is optimized after two calls: one for the forward translation and another for the backward translation. During the forward translation, the encoder takes a sentence $\mathbf{x} = y_{\text{tag}}, x_1, ..., x_M$, where $y_{\text{tag}}$ is the tag of the target language, and samples a translation output $\mathbf{y} = y_1, y_2, ..., y_N$. At each step the decoder randomly picks the next word from the current distribution. This word is fed as input at the next decoding step.

Then, $\mathbf{y}$ is used by the language model RNN trained on the target language. The language model is kept fixed during the whole process. Observe that initially the special begin-of-sentence symbol is fed, so that we can also obtain the probability for the first word. Here, we use the output logits in order to calculate the probability of the sample sentence. The first reward is computed as:

---

[2]https://www.tensorflow.org/tutorials/seq2seq.

$$r_1 = \log P_Y(\mathbf{y}) = \sum_{i=1}^{N} \log P_Y(y_i|y_1, \ldots, y_{i-1}) \tag{5}$$

In addition, the backward translation call is executed: $\mathbf{y}$, plus the source language tag added in the beginning, $x_{\text{tag}}$, is encoded to a sequence of hidden state vectors $h_0, h_1, h_2, \ldots, h_N$ using the encoder. These outputs are passed to the decoder which is fed with the original source words $x_i$ at each step. The output logits are used to calculate the probability of the source sentence $\mathbf{x}$ being reconstructed from $\mathbf{y}$. This value is used as the second reward:

$$r_2 = \log P_\theta(\mathbf{x}|\mathbf{y}) = \sum_{i=1}^{M} \log P_\theta(x_i|x_1, \ldots, x_{i-1}, h_0, h_1, \ldots, h_N) \tag{6}$$

## 4 EXPERIMENTS

We conduct experiments for two language pairs: we evaluate on French↔Spanish in an in-domain setting, and out-of-domain, on English↔French to allow for comparison with results reported in recent literature.

Similar to phrase-based systems we need two kinds of data to train our models: parallel and monolingual. A key factor in the choice of parallel training corpora is the fact that we require parallel data for the supervised language pairs (Z-X, Z-Y), in order to perform zero-shot translation (X-Y). In order to be able to better understand both the performance of the initial model and the effectiveness of our approach, we chose the United Nations Parallel Corpus (UN corpus) (Ziemski et al., 2016). The fully-aligned sub-corpus is composed of sentences from UN documents, which are fully aligned across the six official UN languages, i.e. the same sentence is available in 6 languages. We only use English, Spanish and French.

As the UN corpus is of sufficient size we also use it to collect our monolingual data used for training the language model. Moreover, official development and test sets are provided which comprise 4,000 sentences each, aligned for all the six languages. This allows for experiments to be evaluated, and replicated, in all directions.

In addition, we conduct a second line of experiments on English↔French translation in order to compare our approach with previously reported results. Here, we switch the monolingual data to the News Crawl dataset from WMT14 but keep the parallel UN corpus for initial training,

### 4.1 VOCABULARY

Following Johnson et al. (2016), we tokenize the data using an algorithm similar to Byte-Pair-Encoding (BPE) (Sennrich et al., 2016) in order to efficiently deal with unknown words while still keeping a limited vocabulary.

We calculate the vocabulary on the parallel data sampled for the translation system. We apply 32k merge operations which yields a vocabulary of roughly 33k sub-word units, shared across the three languages. Every model that we train—the language models and the translation systems—relies on this vocabulary. Three language tags are added at the end of the vocabulary in the translation systems for the zero-shot purpose (the language model does not use them).

### 4.2 LANGUAGE MODELS

We sampled 4M random sentences from the parallel UN corpus for each language to train the language models. In other words, we selected 4M Spanish sentences from the union of the Spanish lines contained in the English-Spanish and Spanish-French corpora. Similarly, we collected the French training dataset from the English-French and Spanish-French files. We remove sentences longer than 100 words.

The language model is a 2-layer, 512-unit LSTM, with 65% dropout probability applied on all non-recurrent connections. It is trained using Stochastic Gradient Descent (SGD) to minimize the

|  | **Phrase-based** | **NMT-0** |
|---|---|---|
| Parallel Data | en-fr (1M) en-es (1M) | en-fr (1M) en-es (1M) |
| en→es | 51.10 | 51.93 |
| es→en | 49.55 | 51.58 |
| en→fr | 39.43 | 40.56 |
| fr→en | 40.44 | 43.33 |

Table 1: BLEU scores on the UN corpus test set. The first column refers to a number of baseline phrase-based models, the second is a single multi-lingual NMT model. All models are trained on the same 1M en-es and 1M en-fr aligned sentences, used in both directions. In this setting, NMT-0 is used in the supervised direction.

|  | **NMT-0** | **Dual-0** |
|---|---|---|
| Parallel Data | en-fr (1M) en-es (1M) | en-fr (1M) en-es (1M) |
| Monol. Data |  | es (500k) fr (500k) |
| es→fr | 20.29 | **36.68** |
| fr→es | 19.10 | **39.19** |

Table 2: BLEU scores on the UN corpus test set, using no parallel data for the evaluated direction. The first column (NMT zero-shot, same model as in Table 1) refers to our baseline multilingual model. The Dual-0 model refers to our approach, applying RL with 500k monolingual sentences of each language on top of NMT-0.

negative log probability of the target words. We trained the model for six epochs, halving the learning rate, initially 1.0, in every epoch starting at the third. The norm of the gradients is clipped at 10. We used a batch size of 64 and the RNN is unrolled for 35 time steps. The vocabulary is embedded to 512 dimensions. Each language model was trained on a single Tesla-K80 for approximately 60 hours.

Our Spanish language model obtains a wordpiece-level perplexity of 18.919, while the French one obtains 15.090. Since the perplexities we report are at sub-word unit level, it is not easy to quantify their quality. For that reason, we use the KenLM implementation (Heafield, 2011) with the same sample and vocabulary. We obtain test perplexities of 14.76 and 12.84 for the Spanish and French language models, respectively. While that difference is significant, we do not attempt to improve our language models as they are sufficient to obtain a coarse fluency metric.

### 4.3 Translation systems

For the translation systems, we randomly sampled 1M parallel sentences for each language pair. All sentences are disjoint across language pairs. After sampling, we removed sentences longer than 100 tokens. Development and test sets are used as they are in both cases.

Our base NMT model uses LSTM cells with 1024 hidden units and a 8+1-layer encoder and 8-layer decoder as described by Wu et al. (2016). The baseline zero-shot NMT was trained on two Tesla-P100s for approximately one week.

Training with reinforcement learning is performed on 1M *monolingual* sentences, e.g. 500K each for Spanish and French for the es-fr model. This data does not include aligned pairs across languages. They are also disjoint from the ones used to train the base NMT model. The model configuration is the same as the NMT base model, except for the batch size which needs to be halved for memory resources, given that we now need to load two language models. We use learning rates $\gamma_{1,t} = 0.0002$ and $\gamma_{2,t} = 0.02$ and $\alpha$=0.005, following He et al. (2016). We did not attempt to optimize hyperparameters to fine tune the model. Training takes approximately 3 days on two Tesla-P100s.

## 5 Results

We first report our Spanish↔French results, followed by English↔French.

### 5.1 Supervised performance

We compute the BLEU score on the official test sets, using the `multi-bleu-detok.perl` script, as distributed with Moses (Koehn et al., 2007).[3]

---

[3]`http://www.statmt.org/moses/`

For comparison, the first column of Table 1 reports the BLEU scores for a phrase-based translation system using the same data as our baseline multilingual NMT model. We've build these systems as another baseline using Moses (Koehn et al., 2007) with the common pipeline (MGiza, 5-gram KenLM (Heafield, 2011), Mert) but without including additional monolingual data to make both systems comparable.

The second column reports the scores of our multilingual NMT baseline model (NMT-0). The numbers show that our baseline is a competitive starting point even though the multilingual NMT system learns four translation directions at the same time. We would like to stress, though, that our goal is not to achieve good performance on the supervised directions but to have a reasonable initialization for the reinforcement learning of the unsupervised pairs.

## 5.2 Unsupervised performance

The first column of Table 2 shows the performance of the baseline multilingual NMT model (NMT-0) on the low-resource pair (es-fr), which is the main focus of this work. NMT-0 already obtains a reasonable performance on Spanish and French: 20.29 es→fr and 19.10 fr→es, without having been trained on a single parallel sentence pair. The zero-shot dual system (Dual-0) starts training from the parameters of the NMT-0 model. We observe a marked improvement for the zero-shot translation directions, outperforming the baseline by 16.39 and 20.09 BLEU points for es→fr and fr→es respectively. By inspection, the translation quality seems reasonable.

## 5.3 WMT experiments

In this section we evaluate a zero-shot dual system out of domain, on the WMT14 testset, *newstest2014*, which is the standard testset used in the literature. We train a zero-shot dual system by training on the English-Spanish and Spanish-French UN data. In this setting, English↔French is the zero-shot pair. Training and model setup is as in the previous experiments. The monolingual data for the reinforcement learning for the en-fr pair uses 500k monolingual sentences sampled at random from each language, from the WMT News Crawl corpora, all years, as common in the literature. Training is also kept the same, except for $\gamma_{1,t} = 0.002$ which helped the model converge faster.

Table 3 reports the detokenized BLEU scores computed using sacreBLEU (Post, 2018).

As opposed to the experiments above, the test and RL data are of a different domain from our LM training data (News vs. UN documents). To overcome this mismatch we train English and French language models on 10M random sentences of the News Crawl data. These sentences are disjoint from the RL data. The architecture for the LSTM based LM is mostly kept the same, except for the LSTM size which we increased to 1500 units. As for the training schedule, we train the models for 30 epochs, decreasing the learning rate (initially 1) by 0.8 every epoch after the 8th. We use the same vocabulary as in the previous experiments.

For this experiment we implement additional baselines. First of all, we train single translation direction NMT models for en-es and es-fr on the UN parallel data. These follow the same architecture and training schedule as the multilingual NMT baseline. The Pivoting column in Table 3 refers to calculating the en→fr translation of *newstest2014* by using the models en→es and es→fr in sequence. Similarly, for the other direction. As these single-direction models can allocate more parameters to a translation direction compared to the multi-lingual models, their individual performances are slightly better than the supervised multi-lingual models scores reported in Table 1, leading to reasonably good results, despite the domain mismatch.

The Pseudo NMT baseline instead, uses the Pivoting method to generate back-translations of the monolingual (in-domain) RL data, this way obtaining two synthetic parallel corpora of 500k sentences each. On this data, using the synthetic data as the source, we perform standard supervised training of the NMT system. NMT refers to a supervised en-fr model, trained on UN parallel data. As such, this model is not comparable to the un-supervised models, but shows the significance of domain mismatch between UN and news data.

As above, NMT-0 refers to the zero-shot model trained on en-es and es-fr UN data and Dual-0 is using RL with in-domain data on top of NMT-0.

| | Pivoting | Pseudo NMT | NMT | NMT-0 | Dual-0 | NMT-U |
|---|---|---|---|---|---|---|
| Parallel Data (UN) | es-en (1M) es-fr (1M) | | en-fr (1M) | es-en (1M) es-fr (1M) | es-en (1M) es-fr (1M) | |
| Parallel Synth. (NC) | | en-fr (0.5M) | | | | |
| Monol. Data (NC) | | | | | en (0.5M) fr (0.5M) | Full |
| en→fr | 18.69 | 20.24 | 24.33 | 21.29 | 24.93 | 24.5/ **25.1** |
| fr→en | 16.86 | 21.92 | 24.52 | 22.89 | **25.34** | 23.7/24.2 |

Table 3: Comparison with other approaches. Numbers are BLEU scores on WMT newstest2014. Pivoting refers to translating the testset using the fr-es and es-en single direction NMT models, while Pseudo-NMT refers to the NMT models trained on synthetic data on the source side, obtained by back-translating the RL data using the single direction NMT models. NMT refers to the model trained on parallel UN en-fr data, this is not an unsupervised model. NMT-0 refers to the zero-shot multilingual NMT baseline, trained on en-es and es-fr parallel UN data. Dual-0 reports the results of our approach on top of NMT-0. For NMT-U we cite results from Lample et al. (2018b): the first number refers to the result of the LSTM-based model, while the second is the result of the Transformers-based model. NC refers to the monolingual News Crawl dataset.

On the WMT14 testset we can compare against the results of the unsupervised MT (NMT-U, in Table 3) proposed in (Lample et al., 2018b). Specifically, we report the performance of comparable NMT architectures, both LSTM and Transformers (Vaswani et al., 2017). Dual-0 outperforms the baselines and seems comparable to NMT-U on en→fr, while on fr→en Dual-0 performs best. Lample et al. (2018b) show that phrase-based, or combined phrase-based/neural, methods outperform neural architectures for unsupervised MT (28.11 for en→fr and 27.68 for fr→en. We are investigating ways of including phrase-based information in our zero-shot dual framework.

We notice that Dual-0 outperforms supervised NMT model. Even though the supervised model is trained on parallel en-fr UN data, the testset is out of domain. This means that our approach not only strengthens the zero-shot translation direction, but also works as a domain adaptation approach.

# 6 CONCLUSION

We propose an approach to zero-shot machine translation between language pairs for which there is no aligned data. We build upon a multilingual NMT system (Johnson et al., 2016) by applying reinforcement learning, using only monolingual data on the zero-shot translation pairs, inspired by dual learning (He et al., 2016).

Experiments show that this approach outperforms the multilingual NMT baseline model for unsupervised language pairs, on in-domain evaluations in the UN corpus. In out-of-domain evaluation we show that the zero-shot dual approach performs as well or better than the state of the art for unsupervised MT, for comparable neural architectures, including Transformers.

These results show that this is a promising framework for machine translation for low-resource languages, given that aligned data already exists for numerous other language pairs. This framework seems particularly promising in combination with techniques like bridging, given the abundant data, to and from English and other popular languages. For future work, we would like to understand better the relation between model capacity, number of languages and language families, to optimize information sharing. Exploiting more explicitly cross-language generalization; e.g., in combination with recent unsupervised methods for embeddings optimization (Lample et al., 2018a; Artetxe et al., 2018), seems also promising. Finally, we would like to extend the reinforcement learning approach to include other reward components; e.g., linguistically motivated.

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
