# OpenReview forum: "Zero-shot Dual Machine Translation"
_ICLR.cc/2019/Conference_

### Official Review · AnonReviewer3 · 2018-11-02
**Limited novelty; Experiments are not enough**

**Rating:** 5
**Confidence:** 4

**Review:**


[Summary]
This paper proposed an algorithm for zero-shot translation by using both dual learning (He et al, 2016) and multi-lingual neural machine translation (Johnson et al 2016). Specially, a multilingual model is first trained following (Johnson et al 2016) and then the dual learning (He et al 2016) is applied to the pre-trained model using monolingual data only. Experiments on MultiUN and WMT are carried out to verify the proposed algorithm.

[Details]
1.	The idea is incremental and the novelty is limited. It is a simple combination of dual learning and multilingual NMT.

2.	Many important multilingual baselines are missing. [ref1, ref2]. At least one of the related methods should be implemented for comparison.

3.	The Pseudo NMT in Table 3 should also be implemented as a baseline for MultiUN experiments for in-domain verification.

4.	A recent paper [ref3] proves that using more monolingual data will be helpful for NMT training. What if using more monolingual data in your system? I think using $1M$ monolingual data is far from enough.

5.	What if using more bilingual sentence pairs? Will the results be boosted? What if we use more language pairs?

6.	Transformer (Vaswani et al. 2017) is the state-of-the-art NMT system. At least one of the tasks should be implemented using the strong baseline.

[Pros] (+) A first attempt of dual learning and multiple languages; (+) Easy to follow.
[Cons] (-) Limited novelty; (-) Experiments are not enough.

References
[ref1] Firat, Orhan, et al. "Zero-resource translation with multi-lingual neural machine translation." EMNLP (2016).
[ref2] Ren, Shuo, et al. "Triangular Architecture for Rare Language Translation." ACL (2018).
[ref3] Edunov, Sergey, et al. "Understanding back-translation at scale."EMNLP (2018).

I am open to be convinced.

==== Post Rebuttal ===
Thanks the authors for the response. I still have concerns about this work. Please refer to my comments "Reply to the rebuttal". Therefore, I keep my score as 5.

---

> ### Author Response · Authors · 2018-11-26
> **Response to Reviewer3**
>
> 1. That is correct. Yet, we see the simplicity of the approach as a merit and the results convincing. We are currently working on stabilizing the RL training in order to be able to leverage bigger amounts of monolingual data and our current results (still with the same 1M monolingual sentences) are at 26.36 and 26.04 BLEU for WMT14 en -> fr and fr -> en, which shows the potential of the approach.
>
> 2. Thanks for the pointers! Even though we cannot implement them in this limited time, we will keep them in mind to include it in our research.
>
> 3. We actually did implement them and they were inline with the results in the WMT setting.
>
> 4. We have experimented with more monolingual data (only on the UN data experiments so far) but it does not seem to help as the biggest gains happen at the beginning of the RL training. We are still looking into how to make better use of it.
>
> 5. We have thought about these experiments as well. We definitely expect to obtain a better initial baseline model by using more parallel sentences which, we expect, would boost performance overall. The scenario with more languages also needs to be investigated further, but early conclusions in our experiments show that the performance is consistent.
>
> 6. We don't present Transformer based experiments due to additional implementation effort, which we have yet to tackle. Although Transformer is a state-of-the-art  NMT system, RNN-based models are still competitive. Moreover, we have shown in (1) that RNN-based models can beat Transformers and hope to achieve further improvements.

---

> > ### Comment · AnonReviewer3 · 2018-11-29
> > **Reply to the rebuttal**
> >
> > Thank the authors for the detailed response.
> >
> > [Reply to response #1 and #2]
> > I agree to the point that simple approach can lead to good results. But, considering no related baseline algorithm is implemented, then how could we say we really need such an approach? Hope the authors can address it in the future.
> >
> > Besides, what does the “26.36 and 26.04 BLEU for WMT14 en -> fr and fr -> en,” mean in your response #1? I did not find these two numbers in the paper. Are they the column ``Dual-0’’ in Table 3?
> >
> > [Reply to response #3]
> > I think you should add the results of back-translation for MultiUN settings (do I miss anything) because as pointed by Reviewer 1, the WMT experiments in your paper is not an actual unsupervised setting. Therefore, this baseline should be verified on MultiUN.
> >
> > [Reply to response #4 and #5]
> > I do not find the quantitative results. Please attach them. Besides, why more monolingual data seems not helpful in your setting? How could you leverage more monolingual data more efficiently?
> >
> > [Reply to response #6]
> > What does the ``(1)’’ mean? Can you explain it clearly?

---

> > > ### Author Response · Authors · 2018-11-30
> > > **Reply to Reviewer3**
> > >
> > > Thanks for the quick reply!
> > >
> > > [Reply to #1 and #2]
> > > We believe that pivoting and pseudo-NMTs are a simple, yet competitive baseline. However, we are currently working on adding several additional baselines from the related work.
> > > The numbers you are referring to are not yet included in the paper since the training is still running. We indeed refer to the Dual-0 column in Table 3.
> > >
> > > [Reply to #3]
> > > Agreed. We will add the same sets of results for both experimental settings, making sure that we are using the same setup for both.
> > >
> > > [Reply to #4 and #5]
> > > The reinforcement learning training hits a peak quite fast and after spending some time at this performance level, it deteriorates. That is why we are analyzing how to stabilize the RL training process so we can leverage more monolingual data and potentially improve peak performance. We are happy to add (in the appendix) a learning curve to show how the RL learning progresses. Is this the quantitative result you're looking for?
> > >
> > > [Reply to #6]
> > > We apologize for the confusion. We refer to the first bullet point of the reply, as we found that our latest results (26.36 and 26.04 BLEU for WMT14 en -> fr and fr -> en) exceed those obtained with Transformers from Lample et al. (2018b). That being said one would expect that all numbers could improve when using a large Transformer as the base model and we should try this experiment.

---

### Official Review · AnonReviewer2 · 2018-11-02
**Some nice ideas, but would benefit from a better comparison to related work**

**Rating:** 6
**Confidence:** 3

**Review:**

Pros:
- The paper address the problem of zero-shot translation. The proposed method is essentially to bootstrap a Dual Learning process using a multilingual translation model that already has some degree of zero-shot translation capabilities. The idea is simple, but the approach improves the zero-shot translation performance of the baseline model, and seems to be better than either pivoting or training on direct but out-of-domain parallel data.
- The paper is mostly well written and easy to follow. There are some missing details that I've listed below.

Cons:
- There is very little comparison to related work. For example, related work by Chen et al. [1], Gu et al. [2] and Lu et al. [3] are not cited nor compared against.

Misc questions/comments:
- In a few places you call your approach unsupervised (e.g., in Section 3: "Our method for unsupervised machine translation works as follows: (...)"; Section 5.2 is named "Unsupervised Performance"). But your method is not unsupervised in the traditional sense, since you require lots of parallel data for the target languages, just not necessarily directly between the pair. This may be unrealistic in low-resource settings if there is not an existing suitable pivot language. It'd be more accurate to simply say "zero-shot" (or maybe "semi-supervised") in Section 3 and Section 5.2.
- In Section 3.1 you say that your process implements the three principles outlined in Lample et al. (2018b). However, the Initialization principle in that work refers to initializing the embeddings -- do you pretrain the word embeddings as well?
- In Section 4 you say that the "UN corpus is of sufficient size". Please mention what the size is.
- In Section 4.2, you mention that you set dropout to p=0.65 when training your language model -- this is very high! Did you tune this? Does your language model overfit very badly with lower dropout values?
- In Section 5.2, what is the BLEU of an NMT system trained on the es->fr data (i.e., what is the upper bound)? What is the performance of a pivoting model?
- In Section 5.3, you say you use "WMT News Crawl, all years." Please indicate which years explicitly.
- In Table 3, what is the performance of a supervised NMT system trained on 1M en-fr sentences of the NC data? Knowing that would help clarify the impact of the domain mismatch.
- minor comment: in Section 4.3 you say that you trained on Tesla-P100, but do you mean Pascal P100 or Tesla V100?

[1] Chen et al.: http://aclweb.org/anthology/P17-1176
[2] Gu et al.: http://aclweb.org/anthology/N18-1032
[3] Lu et al.: http://www.statmt.org/wmt18/pdf/WMT009.pdf

---

> ### Author Response · Authors · 2018-11-26
> **Response to Reviewer2**
>
> 1. We'd argue that our approach is unsupervised in the sense that we are training a model without data for the task we are training. From a practical perspective, having parallel data for another language pair is not always possible, as is, to a lesser extent, having monolingual data. We do agree that we should stress that our approach is zero-shot to avoid confusion with unsupervised learning.
>
> 2. We do not pretrain the embeddings on their own; they are trained with the rest of the baseline multilingual model. That is, the zero-shot training (before RL) performs the embedding pre-training as the initialization step. This step initializes not only the embeddings but also the weights of the translation model, which could be considered a stronger variant of the unsupervised lexicon induction via word vectors. Apart from the initialization, our approach also leverages language models and back-translation to learn the zero-shot translation directions in the form of rewards.
>
> 3. We use the bilingually aligned data for each language pair. They are approximately of size 18M, 25M and 22M for en-es, en-fr and es-fr, respectively. Each of the development and test sets contain 4000 sentences.
>
> 4. It is indeed high. We did not tune the hyperparameters of the language model (we use the default ones for the big model in the Tensorflow RNN tutorial) since the weight that we currently use for its reward (the one presented in He et al.) is so low (0.005). However, we plan to analyze its role and we will then need to analyze all hyperparameters more in depth to obtain a better performance.
>
> 5. For a single direction model with the same capacity, es->fr and fr->es achieve 42.50 and 44.86 respectively. These models were trained on the same 1M sentences used for RL (also using their corresponding translations). As for the pivoting experiments on the UN setup, we do not have the exact numbers but they were slightly below the Dual-0 performance.
>
> 6. All the years that include Spanish: 2007-2013.
>
> 7. The NewsCrawl is monolingual and thus, we cannot train a supervised NMT system on it. We did not use the News Commentary corpus, which, while more 'newsy', still differs in style. The Pseudo-NMT is the closest scenario to supervised in-domain training for this setting and we do report these numbers.
>
> 8. We train on NVIDIA Tesla P100 which use the NVIDIA Pascal GPU architecture.

---

### Official Review · AnonReviewer1 · 2018-11-05
**Novelty is not enough. More experiments needed.**

**Rating:** 4
**Confidence:** 5

**Review:**

This paper can be considered as a direct application of dual learning (He et al. (2016)) to the multilingual GNMT model. The first step is to pre-train the GNMT model with parallel corpora (X, Z) and (Y, Z). The second step is to fine-tune the model with dual learning.

1. I originally thought that the paper can formulate the multilingual translation and zero dual learning together as a joint training algorithm. However, the two steps are totally separated, thus the contribution of this paper is incremental.

2. The paper actually used two parallel corpora. In this setting, I suggest that the author should also compare with other NMT algorithm using pivot language to bridge two zero-source languages, such as ``A Teacher-Student Framework for Zero-Resource Neural Machine Translation``. It is actually unfair to compare with the completely unsupervised NMT, because the existence of the pivot language can enrich the information between two zero-resource languages. The general unsupervised NMT is often considered as ill-posed problem. However, with parallel corpus, the uncertainty of two language alignment is greatly reduced, making it less ill-posed. The pivot language also plays the role to reduce the uncertainty.

---

> ### Author Response · Authors · 2018-11-26
> **Response to Reviewer1**
>
> 1. That is correct. We improve a multilingual baseline model using monolingual data for the zero-shot translation. The contribution is, as you point out, incremental. Yet, we believe that the simplicity of the approach makes the results stronger. We have shown in our paper that our approach outperforms various baselines such as the zero-shot multilingual NMT system, pivoting and pseudo-NMTs. Moreover, we show that its performance is in line with similar architectures such as Lample et al. (2018b). We are still working on improving the approach and we have improved our results by 1-2 BLEU so far.
>
> 2. Thank you for the pointer, we will take that work into account. We are aware that our contribution is not totally comparable to fully unsupervised approaches and clearly, introducing a pivot language helps the learning process. However, the approach is unsupervised in the sense that we have no data for the concrete task we're training for which is a setup that we can find for most low resource languages and we believe we should take advantage of it. We'll stress more that we are doing zero-shot to avoid confusion.

---

### Author Response · Authors · 2018-11-26
**Thanks to reviewers**

We'd like to sincerely thank all reviewers for studying our work in such depth and for their detailed and thoughtful comments, which have proven very helpful.

---

### Meta-Review · Area_Chair1 · 2018-12-13
**Reasonable improvements, but novelty incremental**

**Confidence:** 4
**Recommendation:** Reject

**Metareview:**

This paper is essentially an application of dual learning to multilingual NMT. The results are reasonable.

However, reviewers noted that the methodological novelty is minimal, and there are not a large number of new insights to be gained from the main experiments.

Thus, I am not recommending the paper for acceptance at this time.